# Impact of Selected Glucagon-like Peptide-1 Receptor Agonists on Serum Lipids, Adipose Tissue, and Muscle Metabolism—A Narrative Review

**DOI:** 10.3390/ijms25158214

**Published:** 2024-07-27

**Authors:** Zsolt Szekeres, Andras Nagy, Kamilla Jahner, Eszter Szabados

**Affiliations:** 1Department of Laboratory Medicine, Medical School, University of Pecs, 7624 Pecs, Hungary; szekeres.zsolt@pte.hu; 2Faculty of Pharmacy, University of Pecs, 7624 Pecs, Hungary; nagy.andras@pte.hu; 3Department of Medical Imaging, Medical School, University of Pecs, 7624 Pecs, Hungary; jahner.kamilla@pte.hu; 41st Department of Medicine, Division of Preventive Cardiology and Rehabilitation, Medical School, University of Pecs, 7624 Pecs, Hungary

**Keywords:** GLP-1 RA, obesity, serum lipid, adipose tissue, muscle metabolism

## Abstract

Glucagon-like peptide-1 receptor agonists (GLP-1 RA) are novel antihyperglycemic agents. By acting through the central nervous system, they increase satiety and reduce food intake, thus lowering body weight. Furthermore, they increase the secretion of insulin while decreasing the production of glucagon. However, recent studies suggest a more complex metabolic impact through the interaction with various other tissues. In our present review, we aim to provide a summary of the effects of GLP-1 RA on serum lipids, adipose tissue, and muscle metabolism. It has been found that GLP-1 RA therapy is associated with decreased serum cholesterol levels. Epicardial adipose tissue thickness, hepatic lipid droplets, and visceral fat volume were reduced in obese patients with cardiovascular disease. GLP-1 RA therapy decreased the level of proinflammatory adipokines and reduced the expression of inflammatory genes. They have been found to reduce endoplasmic reticulum stress in adipocytes, leading to better adipocyte function and metabolism. Furthermore, GLP-1 RA therapy increased microvascular blood flow in muscle tissue, resulting in increased myocyte metabolism. They inhibited muscle atrophy and increased muscle mass and function. It was also observed that the levels of muscle-derived inflammatory cytokines decreased, and insulin sensitivity increased, resulting in improved metabolism. However, some clinical trials have been conducted on a very small number of patients, which limits the strength of these observations.

## 1. Introduction

A sedentary lifestyle and unhealthy diet increase the incidence of obesity, which has led to a significant increase in the prevalence of type 2 diabetes mellitus (T2DM) [1,2]. Obesity and T2DM are major contributors to morbidity and mortality worldwide [3]. According to the World Health Organization (WHO), obesity has nearly tripled since 1975, as more than 650 million adults were classified as obese in 2016 [4]. It poses a threat to not only the cardiovascular and metabolic system but is also affiliated with certain types of cancer, pneumological, nephrological, skeletal muscle, rheumatologic, dermatologic, and neuropsychologic complications [5]. Obesity is strongly connected to insulin resistance, hypertension, dyslipidemia, and systemic inflammation, which all lead to an elevated risk for both cardiovascular diseases (CVD) and T2DM [6]. The International Diabetes Federation (IDF) reports that approximately 537 million adults aged between 20 and 79 years were diagnosed with diabetes in 2021, which is projected to rise to 643 million by 2030 and 783 million by 2045 [7]. Based on the above data, there is a great need for novel therapeutic drugs that can specifically target weight gain with advantageous metabolic effects. 

Glucagon-like peptide-1 receptor agonists (GLP-1 RAs) have emerged as promising agents that can not only help to achieve glycemic control but also offer additional benefits, including weight reduction and cardiovascular protection as well [8]. GLP-1 is secreted mainly by intestinal enteroendocrine L-cells within the first minutes of nutrient ingestion. GLP-1 receptors (GLP-1Rs) are a type of class B family of 7-transmembrane-spanning, heterotrimeric G-protein-coupled receptors. They are expressed in various tissues, including the lung, heart, pancreas, stomach, intestines, kidneys, muscle, adipose tissue, and even in the nervous system. After activation, GLP-1R activates adenylate cyclase to stimulate cyclic adenosine monophosphate (cAMP) production, thus leading to further secondary messenger pathways [9].

GLP-1 receptor agonists (GLP-1 RAs) are incretin-based agents that mimic the effect of the endogenous incretin hormone GLP-1. They stimulate the release of insulin from pancreatic beta cells, thus improving glycemic control. They reduce hepatic glucose production by suppressing the secretion of glucagon. By slowing gastric emptying, they moderate postprandial glucose spikes. GLP-1 RA also affects the central nervous system, inducing satiety and reducing food intake, thereby leading to significant weight loss [10].

Among GLP-1 RAs, liraglutide (from 1.2 mg/day to 1.8 mg/day subcutaneously) and semaglutide (from 0.25 mg/week to 2.4 mg/week subcutaneously or from 3 mg/day to 14 mg/day orally) are approved not only for the treatment of type 2 diabetes mellitus but also for the treatment of obesity [11].

Semaglutide is a long-acting GLP-1 RA that has a well-known beneficial effect on glucose metabolism, the cardiovascular system, and body weight reduction. It was first approved by the US Food and Drug Administration (FDA) for the treatment of T2DM in 2017. It acts through the same mechanisms as other GLP-1 RAs, but it is distinguished by its efficacy and longer duration of action. As a therapeutic agent, it is available in both oral and injectable forms, thus offering a flexible and effective option for the management of both T2DM and obesity [12].

Liraglutide is another GLP-1 RA that was approved by the FDA in 2010 as an adjunct therapy for the management of T2DM. As the results from clinical trials have shown, it is a potent weight loss agent; thus, the FDA has approved the use of liraglutide for the management of obesity as well. It is also a derivative of GLP-1 that shares a 97% amino acid sequence homology with the original molecule. It is currently available only as a subcutaneous injection [13]. 

The beneficial effects of GLP-1 RA on glucose metabolism and the CV system are well known, and a huge number of studies in the literature are available. In this review, we aimed to summarize the effects of liraglutide and semaglutide on lipid levels, adipose tissue, and muscle metabolism, which are usually less emphasized. We sought to better understand their therapeutic potential beyond their effects on glucose metabolism.

## 2. Methods

Online databases such as PubMed (Available online: pubmed.ncbi.nlm.nih.gov, accessed on 2 June 2024), Google Scholar (Available online: scholar.google.com, accessed on 2 June 2024), and Embase (Available online: embase.com, accessed on 2 June 2024) were used for a comprehensive exploration of the relevant literature. We selected 67 studies (clinical trials, animal models, or meta-analyses) that explored the effects of selected GLP-1 RA (liraglutide and semaglutide) on the lipid profile, adipose and muscle tissues from the years 2010–2024. The key searched terms included GLP-1 RA and lipids, semaglutide and liraglutide and adipose tissue, GLP-1 RA and muscle tissue, semaglutide and liraglutide and sarcopenic obesity. Only original articles, review articles, and meta-analyses published in the English language were included. 

## 3. Discussion

### 3.1. Lipid Profile

Previously, it has been shown that GLP-1 RA therapy affects the blood lipid profile of patients. Hermansen et al. investigated the effect of treatment with liraglutide on postprandial plasma triglyceride concentrations following a standardized high-fat meal. In total, 20 T2DM patients were recruited into a randomized, double-blind, placebo-controlled, cross-over trial. They received 1.8 mg of liraglutide or placebo for 3 weeks before mean postprandial triglyceride and apolipoprotein B48 (ApoB-48) were measured. Both parameters decreased significantly in the GLP-1 RA group. Mean low-density lipoprotein and total cholesterol levels also showed a significant decrease. However, no significant treatment differences were observed for non-esterified fatty acids [14].

To evaluate the effect of semaglutide on the lipid profile, Hjerpsted et al. conducted a randomized, double-blind, placebo-controlled trial. In total, 30 obese patients without diabetes were randomized to once-weekly subcutaneously admitted semaglutide or placebo groups. After the 12-week treatment period, fasting total cholesterol and high-density lipoprotein (HDL) cholesterol levels were lower with semaglutide compared with the placebo, but there was no difference for fasting low-density lipoprotein (LDL) cholesterol. In contrast, fasting concentrations of triglycerides (TGs) and very low-density lipoprotein (VLDL) cholesterol were significantly lower in the treated group. There was no difference between free fatty acids (FFAs) and ApoB-48 when comparing the groups. TG, VLDL, and ApoB-48 levels were all lower after the fat-rich breakfast, but there was no difference in FFA levels [15].

Exposure–response analyses have already shown that the type of administration does not affect the GLP-1 RA treatment response [16]. To assess whether the same beneficial effects on blood lipid levels can be achieved through oral intake, Dahl et al. conducted a randomized, double-blind, single-center, cross-over trial with T2DM subjects. In total, 15 subjects were enrolled in the span of nearly 3 years, of which 13 completed the trial. The cross-over included two treatment periods, one of which included oral semaglutide with a daily dose of 3-7-14 mg escalating, while the other consisted of only placebo tablets. Each period lasted 12 weeks and 3 days, and they were separated by a 5–9 week wash-out period. Lipid metabolism was assessed before and after a fat-rich breakfast. Fasting concentrations of LDL, total cholesterol (TC), triglycerides, VLDL, and ApoB48 were lower in subjects receiving oral semaglutide when compared with the placebo. There was no difference in fasting HDL or FFA between the groups. After the meal, area under the curve 0-8 hour (AUC0-8h) values were significantly lower for TG, VLDL, and ApoB48 in patients taking semaglutide compared to the placebo. However, there was no significant difference in FFA levels between the groups. It was also shown that both fasting and postprandial ApoB48 levels were significantly lower in the semaglutide group when compared with placebo-taking statin users [17].

Moreover, Ghusn et al. assessed the metabolic outcomes of semaglutide through a multicentered retrospective cohort study. Obese patients with or without T2DM were selected and administered semaglutide subcutaneously weekly for more than 3 months. A total of 1023 patients were enrolled in this study. Patients taking semaglutide therapy had a significant decrease in total cholesterol (10.3 mg/dL, n = 132, *p* < 0.001), LDL (5.2 mg/dL, n = 129, *p* = 0.04), and triglycerides (20.4 mg/dL, n = 131, *p* = 0.003). Furthermore, patients diagnosed with non-alcoholic fatty liver disease had a significant decrease in aspartate transferase and alanine transferase as well [18].

Niu et al. investigated the effects of semaglutide on liver injury markers, pro-inflammatory factors, and oxidative stress marker levels in obese mice with ApoB-48 (HFD) induced non-alcoholic fatty liver disease (NAFLD), and 24 mice were randomized into three groups (normocaloric diet (NCD), HFD and Sema). After the administration of semaglutide (30 μmol/kg intraperitoneal), TG, TC, and LDL levels were decreased; however, there was no significant difference between HDL levels. Alanine transaminase (ALT) and aspartate transaminase (AST) levels were also lowered with the treatment. This could be interpreted as the amelioration of dyslipidemia and liver injury. Tumor necrosis factor- α (TNF-α), interleukin-6 (IL-6), and interleukin-1β (IL-1 β) levels were significantly lower in the treated group when compared to the untreated mice. This could mean that the beneficial effect on the liver could be through anti-inflammatory and anti-oxidative stress mechanisms. When observing histology changes, they found that semaglutide-treated mice had reduced lipid droplet percentages when compared to the non-treated group. Semaglutide treatment also decreased the collagen fibrin percentage. It was also shown that semaglutide ameliorated mitochondrial damage, as they were less swollen, and the mitochondrial cristae were relatively more organized when compared to the non-treated group. In summary, these findings theorize that semaglutide is capable of reducing hepatocyte steatosis and fibrosis, lipid deposition, and mitochondrial damage as well [19].

These findings indicate that GLP-1 RA therapy might possess favorable effects on blood lipid profiles and that they may decrease the risk of atherosclerosis and fatty liver disease.

### 3.2. Adipose Tissue

The accumulation of visceral fat is a major component of T2DM. Epicardial adipose tissue (EAT) is the visceral fat of the heart that has its own anatomical, functional, and genetic properties. It develops from brown adipose tissue during embryogenesis. In the adult heart, epicardial fat is primarily located in the atrioventricular and interventricular grooves. Smaller amounts of fat can be also found subepicardially along the walls of the atria and around the two appendages. Patients with T2DM or metabolic syndrome possess higher amounts of EAT independently of traditional body fat indicators. This increased volume of fat progressively fills the space between the ventricles. The amount of EAT can be identified through echocardiography [20]. Iacobellis et al. studied GLP-1 receptors in the EAT. RNA sequencing analysis and polymerase chain reactions (PCRs) were performed to assess the presence of GLP-1 and GLP-2 receptors. Samples were taken from eight subjects with coronary artery disease and type 2 diabetes mellitus while undergoing elective cardiac surgery. Their analysis has shown that EAT expresses both GLP-1R and GLP-2R genes. PCR examination showed that while GLP-1R expression was low, two different sets of intron-spanning primers were detected [21].

For this reason, Iacobellis et al. also performed a 12-week, controlled, parallel study in 80 obese patients with T2DM. The subjects received subcutaneously either semaglutide (up to 1 mg/week) or dulaglutide (up to 1.5 mg/week). The control group consisted of 20 obese TD2M patients receiving metformin therapy. The results have shown that EAT thickness significantly decreased in both the semaglutide and dulaglutide groups after 12 weeks, with a 20% reduction. There was no significant difference in the control group regarding the EAT thickness. Higher doses of semaglutide (1 mg) and dulaglutide (1.5 mg) accounted for a significantly greater reduction in the EAT thickness [22].

The gonadal fat pad is the most studied visceral fat tissue in rodents—which is named epidydimal in males—comparable to the human visceral adipose tissue [23]. Previously, it has been shown that rodent adipocytes express GLP-1 receptors and that liraglutide was able to activate it [24]. Furthermore, GLP-1-R has been found in the epidydimal preadipocytes in vitro and in differentiated adipocytes ex vivo as well [25]. According to Dinghui et al., liraglutide’s effects on lipid metabolism significantly differ from semaglutide regarding its changes in metabolic markers. Semaglutide was shown to attenuate hyperleptinemia more prominently than liraglutide [26]. Since the interaction between adipocytes and semaglutide is not yet fully known, Martins et al. conducted a controlled study on mice to further elaborate on the subject. In total, 40 C57BL/6 male mice were separated into two groups considering the diet: the control group received 10% energy from fat, while the high-fat diet group received 50% energy from fat. After that, each group was further separated into two smaller groups regarding their treatment. Half of the groups received 1.2 µg semaglutide subcutaneously every 3 days, while the control-treated mice received sterile subcutaneous saline for 16 weeks. Fat pad fragments were dissected bilaterally and weighed; then, after preparation, immunohistochemistry, immunofluorescence, and PCR were performed. The results have shown that semaglutide caused a reduction in fat pads when compared to the untreated groups. Plasma cytokine concentrations, which were significantly elevated in the HFD group (leptin, adiponectin, resistin, Monocyte Chemoattractant Protein-1 (MCP-1) and TNF-α) were significantly reduced by semaglutide. HFD increased the expression of inflammatory genes (TNF-α, IL-1β, MCP-1, and leptin), which was ameliorated by semaglutide therapy. Another interesting finding was that while endoplasmic reticulum (ER) stress was augmented in the HFD, which is shown by the elevated expression of the activating transcription factor 4 (ATF4), C/EBP homologous protein (CHOP) and Growth Arrest and DNA Damage 45 (GADD45) genes, which was decreased by semaglutide therapy. Obese mice had hypertrophied adipocytes both in the subcutaneus and epidydimal adipose tissues. Also, macrophage infiltration was observed in the latter adipose tissue, which suggests local inflammation. The cross-sectional area of adipocytes was higher in the HFD group, while semaglutide therapy lowered it. Semaglutide also induced multiloculation and uncoupling protein 1 (UCP1) labeling in the subcutaneous adipose tissue. The expression of genes regarding lipid uptake and lipolysis improved with semaglutide therapy. Semaglutide also improved adipose browning mediators and mitochondrial biogenesis in the subcutaneous adipose tissue [27].

Stafeev et al. previously found that T2DM development is strongly connected to impaired properties of progenitor cells in the adipose tissue. In another study, they aimed to evaluate the effect of semaglutide on adipocyte maturing. In total, 8 T2DM patients with morbid obesity were enrolled. After anthropometric characterization, insulin sensitivity measurement, and blood sample analysis, an open surgical biopsy of abdomen subcutaneous adipose tissue was performed in sterile conditions. After the initial measurements, all patients received subcutaneous injections of semaglutide: 0.25 mg/week in the first month, 0.5 mg/week in the second month, and 1 mg/week from the third to the sixth month. Thereafter, a secondary analysis was performed with the same clinical parameters, and another biopsy was performed, followed by the adipose-derived stem cell isolation. After immunocytochemistry, Western blot, and thermogenesis quantification, they made the following observations. The 6-month therapy led to weight loss and lowered blood glucose levels; however, insulin sensitivity (assessed by both the hyperinsulinemic–euglycemic clamp test and HOMA-IR) was not affected. They also found that the adipose tissue stem cells’ proliferative activity (which is impaired in T2DM patients) was increased after semaglutide therapy. Moreover, the adipogenic potential of the stem cells was also restored after the treatment. The stem cells’ ability to differentiate into lipolytic thermogenic beige adipocytes increased. Moreover, the ADSC-derived adipocytes from T2DM patients after semaglutide therapy consumed glucose more effectively and used it for ATP synthesis without accumulating excessive lipids. This also meant that these adipocytes were able to utilize glucose through uncoupled mitochondrial respiration [28].

These results show that GLP-1 RA therapy can be used not only for the treatment of T2DM and obesity but for the improvement of adipocyte metabolism and function as well. The effects of GLP-1 RA therapy on blood lipids and adipose tissue are summarized in Figure 1.

### 3.3. Muscle Metabolism

#### 3.3.1. The Role of Skeletal Muscle in Glucose and Lipid Metabolism

In the last few decades, in addition to the adipose tissue, skeletal muscle has also been described as an endocrine organ that produces more than 650 myokines. Skeletal muscles communicate, with other organs, such as adipose tissue, liver, pancreas, bones, and brain with these myokines, while some myokines act within the muscle [29]. Skeletal muscle is involved in glucose and lipid metabolism, and its insulin resistance plays an important role in the pathogenesis of diabetes [30].

The best-studied myokine is IL-6. IL-6 has been shown to stimulate β-cell proliferation in vivo and prevent metabolic stress-induced apoptosis [31]. Furthermore, IL-6 increases basal and insulin-stimulated glucose uptake in vitro and in healthy humans in vivo [29]. Ellingsgaard et al. previously showed that an acute increase in IL-6 stimulated GLP-1 secretion in both the intestinal L-cells and pancreatic β-cells [32]. A recent study investigated the effects of IL-6 on postprandial glycemia and insulin secretion in humans and found that IL-6 delayed the rate of gastric emptying [33], which is a major regulator of postprandial glucose levels [34]. In vitro and in vivo studies show that muscle-derived IL-6 enhances lipolysis and fat oxidation through a mechanism involving AMPK activation [35] and that IL-6 autoantibodies appear to be involved in the pathogenesis of type 2 diabetes [36].

Excessive adipose tissue produces a number of proinflammatory cytokines, such as TNF-α, IL-6, and IL-1β, promoting local and systemic chronic low-grade inflammation and resulting in insulin resistance. Fat cell infiltration of the muscles inhibits the differentiation of muscle fibers, changes the protein metabolism pathway, activates the protein degradation system, and subsequently leads to muscle degeneration [37,38].

When obese individuals successfully lose weight, their muscle mass usually decreases as their body fat decreases [39]. Since skeletal muscle is the main site of glucose utilization, and a decrease in skeletal muscle may be associated with impaired glucose metabolism. Skeletal muscle accounts for approximately 40–45% of oral glucose and 80–85% of insulin-mediated glucose uptake. Consequently, a decrease in muscle mass may lead to an increase in plasma glucose [40]. In addition, skeletal muscle loss is associated with an increased risk of sarcopenia and frailty, particularly in diabetic and elderly patients [41]. Therefore, it is advisable to predominantly reduce fat without significant muscle loss.

#### 3.3.2. GLP-1 RA and Muscle Mass and Function

Some recent reports suggest that GLP-1 RA can reduce body weight without significantly reducing muscle mass. In the Liraglutide Effect and Action in Diabetes Trial (LEAD-3), treatment with the GLP-1 RA liraglutide significantly reduced body fat, but there was no significant reduction in lean body mass [42]. Perna et al. reported that obese elderly patients treated with liraglutide for 24 weeks had a significant reduction in fat mass but not in appendicular lean mass [43]. Blundell et al. investigated the effect of semaglutide on obese patients. After 12 weeks of treatment, the average body fat and lean mass decreased by 3.5 kg and 1.1 kg, respectively [44]. Ozeki et al. reported that semaglutide effectively reduced body fat while maintaining muscle mass in obese type 2 diabetic patients [45]. 

Tanaka et al. found that oral semaglutide improved glycemic control, while BMI and total body fat decreased, but total body lean mass and skeletal muscle index remained unchanged [46].

Volpe et al. also found that once-weekly semaglutide injections significantly reduced body fat, with no significant changes in muscle mass and strength [47]. Later, they also investigated the effect of oral semaglutide, which, in addition to improving blood sugar and body weight, has a beneficial effect on body composition by reducing fat mass and preserving muscle mass, with better overall fluid distribution [48]. 

#### 3.3.3. GLP-1 RA on Muscle Atrophy

Several reports have demonstrated the molecular role of GLP-1 in skeletal muscle homeostasis and the therapeutic potential of GLP-1 RA in skeletal muscle atrophy [49].

Skeletal muscle is made up of different types of muscle fibers, and obesity can lead to fiber-type switching [50]. Ren and colleagues observed a decrease in the ratio of type I/II muscle fibers in a group of mice fed with a high-fat diet, suggesting that obesity led to a decrease in oxidative muscle fibers and an increase in glycolytic muscle fibers, which could exacerbate metabolic disorders. After treatment with semaglutide, the proportion of muscle fibers was reversed [51].

In another study, Ren et al. observed that semaglutide improved sarcopenic adiposity in obese mice. TNF-α, IL-6, IL-1β, and HOMA-IR were significantly lower in the semaglutide-treated group, suggesting that semaglutide has a beneficial effect on muscle function by reducing the inflammatory response and insulin resistance [51].

Patients with obesity often experience muscle atrophy [52]. Muscle protein degradation is mainly mediated via the ubiquitin–proteasome and autophagy–lysosome systems [53]. Atrogin-1, a muscle-specific ubiquitin ligase, plays a key role in muscle atrophy [54]. On the contrary, muscle-derived regulatory factor myogenin is pivotal in the promotion of muscle recovery [55]. Sirtuin-1 (SIRT-1) is an important regulator of muscle metabolism [56]. Recent studies have shown that the activation of SIRT1 plays a crucial role in the prevention of age-related muscle atrophy [57]. The activation of SIRT1 may also help improve certain skeletal muscle diseases [58]. Xiang et al. found that in the group treated with palmitoic acid, the expression of muscle atrophy markers (Atrogin-1) of C2C12 myotube cells increased, and the expression of myogenic differentiation markers (MyoD, Myogenin) and SIRT1 decreased, which significantly improved treatment with liraglutide and semaglutide [59]. These results suggest that liraglutide and semaglutide may improve skeletal muscle atrophy by activating SIRT1.

It has also been suggested that GLP-1 RA can increase muscle mass, muscle fiber size, and muscle function by inhibiting the expression of myostatin and muscular dystrophy factor and improve muscle atrophy by upregulating myostatin through a GLP1R-mediated signaling pathway [60].

Mitochondria are a major part of energy production, and the number of mitochondria can directly affect muscle function [61]. Semaglutide significantly improved muscle fiber structure, increased the number of mitochondria, and improved muscle function in obese mice [51]. A recent study investigated the potential therapeutic benefits of GLP-1 RA on oxidative stress, mitochondrial respiration leukocyte–endothelial interactions, inflammation, and carotid intima-media thickness (CIMT) in T2D patients. They found that GLP-1 RA treatment improved redox status and mitochondrial respiration, and reduced leukocyte–endothelial interactions, inflammation, and carotid intima-media thickness in diabetic patients, potentially reducing the risk of atherosclerotic processes [62].

#### 3.3.4. Effects of GLP-1 RA on Fatty Acid and Amino Acid Metabolism in Myocytes

Possible metabolic factors for the decrease in muscle mass during obesity are an abnormal metabolism for fats and organic acids in muscle tissues. Ectopic accumulation of lipids in tissues and incomplete mitochondrial long-chain fatty acid (LCFA) β-oxidation are major features of insulin resistance and T2DM [63]. LCFAs can lead to inflammatory responses, especially arachidonic acid (AA), as a bridge connecting lipid metabolism with immunity and inflammation and have clear anti-inflammatory effects [63]. The main function of long-chain acylcarnitines in the muscle is to ensure the transport of long-chain fatty acids into the mitochondria. Previous studies have shown that patients with type 2 diabetes have increased levels of long-chain acylcarnitines, which can activate proinflammatory signaling pathways such as COX2, JNK, and ERK pathways, which can increase inflammatory processes [64]. Ren and colleagues found that in obese mice, semaglutide reduced glycerophospholipid and long-chain fatty acid levels, which alleviated inflammatory processes [51]. They also found high concentrations of medium-chain fatty acids, which have previously been shown to promote lipid catabolism and stimulate thermogenesis in brown adipose tissue [65]. 

Skeletal muscle is the main site of amino acid metabolism and plays an essential role in protein synthesis, muscle structure, and the maintenance of muscle function. A decrease in their level directly affects protein synthesis, which leads to a decrease in muscle mass [66]. In obese mice, organic acids, such as histidine, isoleucine, proline, valine, and phenylalanine, were found to decrease, but semaglutide can significantly increase organic acid levels [67]. Among them are the branched-chain amino acids isoleucine and valine, which can reduce muscle breakdown and promote muscle protein synthesis. Appropriate branched-chain amino acids also enhance lipid oxidation and lipogenesis, maintaining normal lipid metabolism in the muscles [68]. Semaglutide attenuated skeletal muscle atrophy is associated with chronic liver disease in DDC-fed (diethoxycarbonyl-1,4-dihydrocollidine) diabetic mice. In this model, semaglutide exerted protective effects on liver inflammation, fibrosis, and ROS accumulation, as well as the activated GLP-1R signaling pathway, leading to the inhibition of ubiquitin–proteasome system (UPS)-mediated proteolysis and to the promotion of myocyte myogenesis [69].

Human vascular endothelial cells express abundant GLP-1 receptors as well as insulin receptors [70]. Wang et al. recently reported that GLP-1 RA infusion increased microvascular blood flow in skeletal muscle and myocardium by approximately 30% and 40%, respectively, in obese human subjects [71]. These effects of GLP-1 RA can increase the delivery of oxygen, nutrients, and hormones, such as insulin, to myocytes, which is important for maintaining muscle mass and function. A previous study also reported that GLP-1 RA directly induced myogenesis through a cAMP-dependent complex network in rodent skeletal muscle [49]. In addition, previous reports have shown that liraglutide directly activates glucose transport in rat and mouse skeletal muscle cells [72]. 

The possible beneficial effects of GLP-1 RA are summarized in Figure 2.

## 4. Future Perspectives

The efficacy of GLP-1 RA and the overall positive benefits for weight loss are remarkable, but the challenge of long-term maintenance in weight loss remains unsolved [73]. The STEP-1 extension trial of semaglutide, for instance, in obese adults reported that individuals regained roughly two-thirds of the weight they had lost during the 52-week follow-up period after stopping the treatment [74]. Fat loss during dieting or GLP-1 RA treatment is associated with varying degrees of muscle mass loss [75]. Skeletal muscle is an important participant in insulin action and glucose homeostasis and contributes significantly to energy expenditure [76]. Reduced energy expenditure is thought to promote weight gain [77]. Therefore, a weight loss strategy that preserves muscle mass provides long-term weight maintenance with metabolic and functional benefits [75]. Activin type 2 receptor (ActRII) ligands such as myostatin and activin A are known to negatively regulate muscle size, and the ActRII blockade not only increases muscle size but also decreases fat mass [78]. Combining ActRII blockade with bimagrumab and the GLP-1 receptor agonism may be associated with improved body composition and metabolic health [73]. Furthermore, another study is recruiting patients with non-alcoholic fatty liver disease to receive semaglutide therapy to evaluate hepatic response to oral glucose intake [79].

## 5. Conclusions

Several studies demonstrated the beneficial effects of GLP-1 RA in type 2 diabetes mellitus. In a large systematic review and meta-analysis, the CV effects of different GLP1-RAs were evaluated in patients with type 2 diabetes mellitus, and it was found that major cardiac events, cardiovascular death, myocardial infarction, and stroke were significantly reduced [80].

GLP-1 RA agents affect metabolism on multiple different levels. In our review, we aimed to summarize the effects of certain GLP-1 RAs, liraglutide and semaglutide, on serum lipid levels, visceral adipose tissue, and muscle tissue metabolism. Liraglutide and semaglutide are successfully used in weight reduction. In addition, most of the studies proved their beneficial effects on serum lipid levels, such as TC, LDL, VLDL, and TG. Diet-induced adipocyte alterations were also restored with GLP-1 RA therapy, resulting in proper adipocyte function and metabolism. They reduce the amount of visceral fat and decrease adipose infiltration of the skeletal muscles and epicardial adipose tissue.

In preclinical and small clinical studies, GLP-1 RA therapy increased the microvascular blood flow of muscle tissues, decreased muscle atrophy, improved myocyte function and metabolism, and decreased myocyte-derived inflammatory cytokines and insulin resistance, leading to favorable muscle metabolism and body composition. 

However, it should be noted that a significant part of these clinical studies was performed on a small number of patients, which limits their results. Further studies are needed to confirm these findings.

## Figures and Tables

**Figure 1 ijms-25-08214-f001:**
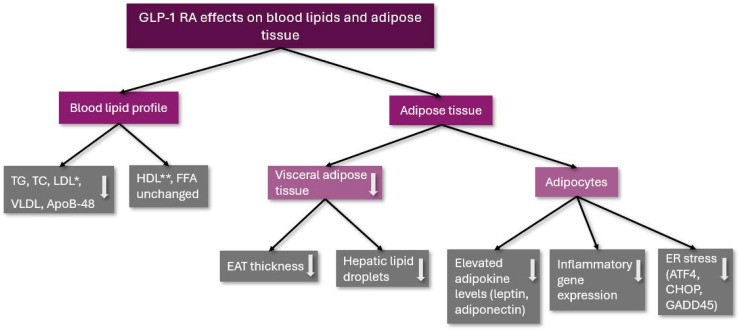
The effects of GLP-1 RA on blood lipids and adipose tissue. TG: triglycerides; TC: total cholesterol; LDL: low-density lipoprotein cholesterol; VLDL: very low-density lipoprotein cholesterol; ApoB-48: Apolipoprotein B-48; HDL: high-density lipoprotein cholesterol; FFA: free fatty acids; EAT: epicardial adipose tissue; ER: endoplasmic reticulum; ATF4: activating transcription factor 4; CHOP: C/EBP homologous protein (CHOP); GADD45: Growth Arrest and DNA Damage 45; *: remained unchanged in a clinical trial; and **: was decreased in a clinical trial; ↓: decreased.

**Figure 2 ijms-25-08214-f002:**
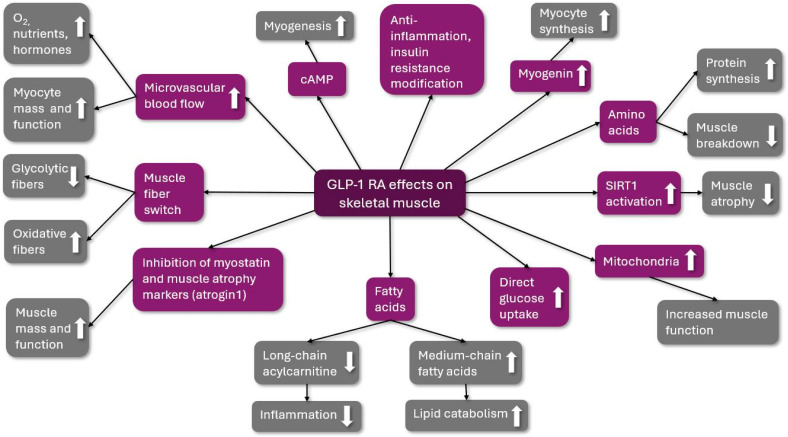
The possible beneficial effects of GLP-1 RA on skeletal muscle. GLP-1 RA: glucagon-like peptid-1 receptor agonist; O_2_: oxygen; cAMP: cyclic adenosine monophosphate; and SIRT1: Sirtuin 1; ↑: increased; ↓: decreased.

## Data Availability

Not applicable.

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
