# Peer review of "Impact of Selected Glucagon-like Peptide-1 Receptor Agonists on Serum Lipids, Adipose Tissue, and Muscle Metabolism—A Narrative Review"

_ijms, 2024, doi:10.3390/ijms25158214_

Round 1

Reviewer 1 Report

Comments and Suggestions for Authors

Reviewer comments are attched

Comments on the Quality of English Language

Author Response

Answer to Reviewer 1

Dear Reviewer,

Thank you very much indeed for your time and efforts reviewing our manuscript! We are grateful for your useful observations and remarks. We did our best answering your questions and hope that the revised version will be eligible for you.

Comment (1)

 In the title, the word “GLP-1 agonists” throughout the main text narrowed down exclusively to “semaglutide”, a one prescribed medication and a few others in some sections. GLP-1 agonists are a whole class comprising several other long-acting and short-acting medications. Suggestion is either to specify the “semaglutide” or the word “selected GLP-1 agonists” in the title or expand the main text by adding similar studies of other GLP-1 agonists with names, as the main text could confuse the reader.

Answer:

Thank you for the suggestion! We have specified the title with “selected” and slightly changed the whole title:

Impact of selected GLP-1 agonists on Serum Lipids, Adipose Tissue, and Muscle Metabolism – a narrative review (Page 1, 2-3)

Comment (2)

In the abstract, most of the sentences are in first person and the abstract seems rather like a research abstract, the suggestion is to use the third person or refer to the source of constituent sentences.

Answer:

Thank you for the suggestion! We have modified the abstract according to your comments:

It has been found that GLP-1 RA therapy was associated with decreased serum cholesterol levels. (Page 1, 18-19)

They have been found to reduce endoplasmic reticulum stress in adipocytes, leading to better adipocyte function and metabolism. (Page 1, 22-23)

It was also observed that the levels of muscle derived inflammatory cytokines decreased, and insulin sensitivity increased resulting in improved metabolism. (Page 1, 25-27)

Comment (3)

In the introduction section, the authors started with diabetes (para#1), switched shortly to cardiovascular disorders, and then linked these two metabolic anomalies with obesity in the context of cardiometabolic health. The integration of two leading disorders with obesity (the master theme of the study) needs a current state of knowledge and uncertainties. Taking advantage of the available literature, the authors need to articulate what and why the intricated relationships of obesity are important to review in the context of the two metabolic disorders.

Answer:

Thank you for the suggestion! We added relevant studies to emphasize the relationship between obesity and T2DM and CVDs and changed this paragraph thoroughly:

A sedentary lifestyle and unhealthy diet increased the incidence of obesity, which has led to a significant increase in the prevalence of type 2 diabetes mellitus (T2DM) [1-2]. Obesity and T2DM are major contributors to morbidity and mortality worldwide [3]. According to the World Health Organization (WHO) obesity has nearly tripled since 1975, as more than 650 million adults were classified as obese in 2016 [4]. It poses a threat to not only the cardiovascular and metabolic system, but it is also affiliated with certain types of cancer, pneumological, nephrological, skeletal muscle, rheumatologic, dermatologic and neuropsychologic complications too [5]. Obesity is strongly connected to insulin resistance, hypertension, dyslipidemia and systemic inflammation that all lead to an elevated risk for both cardiovascular diseases (CVD) and T2DM [6]. The International Diabetes Federation (IDF) reports that approximately 537 million adults aged between 20 and 79 years were diagnosed with diabetes in 2021, which is projected to rise to 643 million by 2030 and 783 million by 2045 [7]. Based on the above data, there is a great need for novel therapeutic drugs that can specifically target weight gain with advantageous metabolic effects.  (Page 2, 34-48)

Comment (4)

In methods, section two is too simplified. It needs to be clarified how many studies were included, how many years were considered, and the logical criteria of selected studies. Also, the straight description is confusing. The section needs a flow diagram specifying the studies selected, database coverage, inclusion/exclusion criteria, and study design.

Answer:

Thank you for your comment! We added more details and description to our methods section. However, unfortunately we can’t provide a flow chart to this section, since this is a narrative review, and for instance, we did not have exclusion criteria. Relevant studies were involved since 2010.

We have selected 67 studies (clinical trials, animal models or meta-analyses) that explored the effects of selected GLP-1 RA (liraglutide and semaglutide) on the lipid profile, adipose and muscle tissues from the years 2010-2024. The key searched terms included: GLP-1 RA and lipids, semaglutide and liraglutide and adipose tissue, GLP-1 RA and muscle tissue, semaglutide and liraglutide and sarcopenic obesity. (Page 3, 90-94)

Comment (5)

In the discussion section, the authors claim an extensive review of the latest clinical trial as mentioned in the method section (line# 100-101); however, I noticed the same outcomes are concluded after each study. Example, in the lipid profile 3.1 section: the only outcome considered is improving the lipid profile. The point is there are always claims and counterclaims, did the author find any studies in contradiction to add limitations to your observations or make a case on lack of such literature in the review where appropriate. This may be applied to other sections too!

Answer:

Thank you for your suggestion! We have included counterarguments in the text and in Figure 1 as well:

After the 12-week treatment period, fasting total cholesterol and high-density lipoprotein (HDL) cholesterol levels were lower with semaglutide compared with placebo, but there was no difference for fasting low-density lipoprotein (LDL) cholesterol. (Page 3, 110-113)

Figure 1: *: remained unchanged in a clinical trial; **: was decreased in a clinical trial (Page 6, 256)

Comment (6)

Generally, cardiometabolic health varies from region to region and population to population based on genetic, epigenetic, and social determinants. I noticed that each parameter discussed is only96 taken from a single clinical study as cited 16, 17, 18, 32, and so on. My first observation is that the number of patient studies seems very less such as 20 DM patients in the first para of the lipid profile section, then 30 DM-obese patients in consecutive para#2, the next 13 patients, and then pre-clinical findings for inflammatory parameters. The studies discussed are rather acute and speculations and I am not convinced with the author's claim based on few studies. Furthermore, the authors linked a few studies; however, an exclusive clinical outcome cannot be concluded with only one article performed of one region. Did the authors try to connect the same clinical studies from different regions with different populations to reach a solid conclusion.

Answer:

Thank you for your comment! We included a novel literature with greater patient population to support our conclusions.

Moreover, Ghusn et al. assessed the metabolic outcomes of semaglutide through a multicentered retrospective cohort study. Obese patients with or without T2DM were selected, who administered semaglutide subcutaneously weekly for more than 3 months. 1023 patients were enrolled into this study. Patients taking semaglutide therapy had a significant decrease in total cholesterol (10.3 mg/dL, n = 132, p < 0.001), LDL (5.2 mg/dL, n = 129, p = 0.04), and triglycerides (20.4 mg/dL, n = 131, p = 0.003). Furthermore, patients diagnosed with non-alcoholic fatty liver disease, had a significant decrease in aspartate transferase and alanine transferase as well. (Page 3, 136- Page 4, 143.)

But of course, we agree with the reviewer that many studies has a very small number of patients, so we refer to this in some places in the article:

However, some of the clinical trials have been conducted on a very small number of patients, which limits the strength of these observations. (Page 1, 27-28)

However, it should be noted that significant part of these clinical studies was per-formed on a small number of patients, which limits their results. Further studies are needed to confirm these findings. (Page 10, 431-433)

Comment (7)

There are also lean or non-obese diabetic patients. Did the authors try to point out such clinical studies and observe the effects of GLP-1 agonists on the targeted parameters in those patients as an additional comparator?

Answer:

Thank you for your comment! We have only included T2DM and obese patients into our narrative review as up to date there are not enough studies evaluating the effects on body composition and lipid profile in lean patients to make proper assumptions. It would be interesting to assess the effects of GLP-1 RA in lean patients in the future.

Comment (8)

The end of each section in the discussion lacks a conclusive remark. The suggestion is to add clinical significance or implications to each section 3.1, 3.2, and 3.3 subsections at the end to convince the readers.

Answer:

Thank you for your suggestion! We have added a summary to each section.

These findings indicate that GLP-1 RA therapy might possess favorable effects on blood lipid profile and that they may decrease the risk of atherosclerosis and fatty liver disease. (Page 4, 163-165)

These results show that GLP-1 RA therapy can be used not only for the treatment of T2DM and obesity, but for the improvement of adipocyte metabolism and function as well. (Page 6, 247-249)

Comment (9)

The information in the figure legends could be expanded to make it stand alone. An open option is that homogeneous and heterogeneous clinical studies can be tabulated to summarize the main findings. Adding tables may strengthen the current review.

Answer:

Thank you for your suggestion! We were given a relatively short time to revise the article, so we could not prepare the suggested table, although we agree that a tabular summary helps the reader to quickly understand the topic.

Comment (10)

Is it possible to do a web search such as NIH clinical trial and provide a table to gather some useful ongoing studies on targeted parameters linked to cardiometabolic health to support the extracted clinical outcomes from previous studies?

Answer:

Thank you for your suggestion! After a search on NIH clinical trials we were able to identify one ongoing study that is still in the recruitment phase regarding the effects of semaglutide on non-alcoholic fatty liver disease:

Furthermore, another study is recruiting patients with non-alcoholic fatty liver disease receiving semaglutide therapy, to evaluate the hepatic response to oral glucose intake [80]. (Page 10, 411-412)

Comment (11)

Future perspectives can be improved by your wording in addition to most of the perspectives you mentioned of others.

Answer:

Thank you for your suggestion! We have changed this paragraph thoroughly:

The efficacy of GLP-1 RA and the overall positive benefits for weight loss are remarkable, but the challenge of a long-term maintenance of weight loss remains unsolved [73]. The STEP-1 extension trial of semaglutide for instance in obese adults reported that individuals regained roughly two-thirds of the weight they had lost during the 52-week follow-up period after stopping the treatment [74]. Fat loss during dieting or GLP-1 RA treatment is associated with varying degrees of muscle mass loss [75]. Skeletal muscle is an important participant of insulin action and glucose homeostasis and contributes significantly to the energy expenditure [76]. Reduced energy expenditure is thought to promote weight gain [77]. Therefore, a weight loss strategy that preserves muscle mass provides long-term weight maintenance with metabolic and functional benefits [75]. Activin type 2 receptor (ActRII) ligands such as myostatin and activin A are known to negatively regulate muscle size, and ActRII blockade not only increases muscle size but also decreases fat mass [78]. Combined ActRII blockade with bimagrumab and GLP-1 receptor agonism may be associated with improved body composition and metabolic health [73]. Furthermore, another study is recruiting patients with non-alcoholic fatty liver disease receiving semaglutide therapy, to evaluate the hepatic response to oral glucose intake [80]. (Page 9, 397- Page 10, 412)

Reviewer 2 Report

Comments and Suggestions for Authors

Authors collected the available literature to give a comprehensive summary of their effects on serum lipids, adipose and muscle tissues in this review article. Please conduct the concerns below.

1.      Source of the used literatures need to introduce in detail.

2.      Rationale to target on lipid profiles reminded obscure.

3.      In line 127, “Exposure-response analyses have already shown that the type of administration does not affect the GLP-1 RA treatment response [19]”. It was an abstract only and details were required.

4.      The potential of GLP-1R agonists (RA) in skeletal muscle atrophy needs to link with clinical applications.

5.      The effective dose of GLP-1 RA for each disorder was not conducted in clear.

6.      The use of semaglutide belonged to major in current report. Why?

7.      The merits in weight loss seem not enough for GLP-1 RA.

8.      Limitation(s) may strengthen this report.

Author Response

Answer to Reviewer 2

Dear Reviewer,

Thank you very much indeed for your time and efforts reviewing our manuscript! We are grateful for your useful observations and remarks. We did our best answering your questions and hope that the revised version will be eligible for you.

Comment (1)

 Source of the used literatures need to introduce in detail.

Answer:

Thank you for your comment! We added more details and description to our methods section.

We have selected 67 studies (clinical trials, animal models or meta-analyses) that explored the effects of selected GLP-1 RA (liraglutide and semaglutide) on the lipid profile, adipose and muscle tissues from the years 2010-2024. The key searched terms included: GLP-1 RA and lipids, semaglutide and liraglutide and adipose tissue, GLP-1 RA and muscle tissue, semaglutide and liraglutide and sarcopenic obesity. (Page 3, 90-94)

Comment (2)

Rationale to target on lipid profiles reminded obscure.

Answer:

Thank you for your comment! We have added more details regarding the effects on the lipid profile changes:

These findings indicate that GLP-1 RA therapy might possess favorable effects on blood lipid profile and that they may decrease the risk of atherosclerosis and fatty liver disease. (Page 4, 163-165)

In a large systematic review and meta-analysis, the CV effects of different GLP1-RA were evaluated in patients with type 2 diabetes mellitus, and it was found that major cardiac events, cardiovascular death, myocardial infarction and stroke were significantly reduced. (Page 10, 415-418)

Comment (3)

In line 127, “Exposure-response analyses have already shown that the type of administration does not affect the GLP-1 RA treatment response [19]”. It was an abstract only and details were required.

Answer:

Thank you for your suggestion! We added more information regarding the reference!

  1. Rune, V.O.; Andrea, N.; Steen, H.I.; Tine, A.B.; Rasmus, J.K. Clinical Pharmacokinetics of Oral Semaglutide: Analyses of Data from Clinical Pharmacology Trials, Clin Pharmacokinet. 2021, 60(10), 1335-1348. doi: 10.1007/s40262-021-01025-x. (Page 11, 481-482)

Comment (4)

The potential of GLP-1R agonists (RA) in skeletal muscle atrophy needs to link with clinical applications.

Answer:

Thank you for your comment! We added more details regarding the clinical applications.

In addition, skeletal muscle loss is associated with an increased risk of sarcopenia and frailty, particularly in diabetic and elderly patients [41]. Therefore, it is advisable to predominantly reduce fat without significant muscle loss. (Page 286-289)

Comment (5)

The effective dose of GLP-1 RA for each disorder was not conducted in clear.

Answer:

Thank you for your comment! We have added the doses of semaglutide and liraglutide as well:

Among GLP-1 RA, liraglutide (from 1.2 mg/day to 1.8 mg/day subcutaneously) and semaglutide (from 0.25 mg/week to 2.4 mg/week subcutaneously, or from 3 mg/day to 14 mg/day orally) are approved not only for the treatment of type 2 diabetes mellitus but also for the treatment of obesity [11].

(Page 2, 65-68)

Comment (6)

The use of semaglutide belonged to major in current report. Why?

Answer:

Thank you for your comment! We included liraglutide and semaglutide in our review, since they are FDA approved for the treatment of obesity. We have changed our manuscript to include more information regarding liraglutide:

Impact of selected GLP-1 agonists on Serum Lipids, Adipose Tissue, and Muscle Metabolism – a narrative review (Page 1, 2-3)

Liraglutide is another GLP-1 RA that was approved by the FDA in 2010 as an adjunct therapy for the management of T2DM. As the results from clinical trials have shown, it is a potent weight loss agent, thus the FDA has approved the use of liraglutide for the management of obesity as well. It is also a derivative of GLP-1 that shares 97% aminos acid sequence homology with the original molecule. It is currently available only as subcutaneous injection [13].  (Page 2, 76-81)

Comment (7)

The merits in weight loss seem not enough for GLP-1 RA.

Answer:

Thank you for your comment! We focused on GLP-1 receptor agonists that are FDA approved for the treatment of obesity and on studies that involved obese patients. We have added more information regarding the weight loss effect of these agents:

They reduce hepatic glucose production by suppressing the secretion of glucagon. By slowing gastric emptying, they moderate postprandial glucose spikes. GLP-1 RA also affect the central nervous system, inducing satiety and reducing food intake, thereby leading to significant weight loss [10]. (Page 2, 61-64)

Among GLP-1 RA, liraglutide (from 1.2 mg/day to 1.8 mg/day subcutaneously) and semaglutide (from 0.25 mg/week to 2.4 mg/week subcutaneously, or from 3 mg/day to 14 mg/day orally) are approved not only for the treatment of type 2 diabetes mellitus but also for the treatment of obesity [11].

(Page 2, 65-68)

Semaglutide is a long-acting GLP-1 RA that has a well-known beneficial effect on glucose metabolism, cardiovascular system and body weight reduction. It was first approved by the US Food and Drug Administration (FDA) for the treatment of T2DM in 2017. It acts through the same mechanisms as other GLP-1 RA, but it is distinguished by its efficacy and longer duration of action. As a therapeutic agent it is available in both oral and injectable forms thus offering a flexible and effective option for the management of both T2DM and obesity [12]. (Page 2, 69-75)

As the results from clinical trials have shown, it is a potent weight loss agent, thus the FDA has approved the use of liraglutide for the management of obesity as well. (Page 2, 77-79)

Comment (8)

Limitation(s) may strengthen this report.

Answer:

Thank you for your suggestion! We have added limitations to the end of the abstract and to the end of the conclusions.

However, some of the clinical trials have been conducted on a very small number of patients, which limits the strength of these observations. (Page 1, 27-28)

However, it should be noted that significant part of these clinical studies was performed on a small number of patients, which limits their results. Further studies are needed to confirm these findings. (Page 10, 431-433)